# “In Initiative Overload”: Australian Perspectives on Promoting Physical Activity in the Workplace from Diverse Industries

**DOI:** 10.3390/ijerph16030516

**Published:** 2019-02-12

**Authors:** Josephine Y. Chau, Lina Engelen, Tracy Kolbe-Alexander, Sarah Young, Heidi Olsen, Nicholas Gilson, Nicola W. Burton, Adrian E. Bauman, Wendy J. Brown

**Affiliations:** 1Prevention Research Collaboration, Sydney School of Public Health and Charles Perkins Centre, University of Sydney, Sydney 2006, Australia; lina.engelen@sydney.edu.au (L.E.); sarah.young@hnehealth.nsw.gov.au (S.Y.); Adrian.bauman@sydney.edu.au (A.E.B.); 2Department of Health Systems and Populations, Faculty of Medicine and Health Sciences, Macquarie University, Sydney 2109, Australia; 3School of Human Movement and Nutrition Sciences, The University of Queensland, Brisbane 4072, Australia; Tracy.Kolbe-Alexander@usq.edu.au (T.K.-A.); heidi.olsen@griffithuni.edu.au (H.O.); n.gilson1@uq.edu.au (N.G.); n.burton@griffith.edu.au (N.W.B.); wbrown@uq.edu.au (W.J.B.); 4School of Health and Wellbeing, University of Southern Queensland, Ipswich 4305, Australia; 5School of Applied Psychology, Griffith University, Mt Gravatt, Queensland 4122, Australia

**Keywords:** physical activity, workplace, health promotion, qualitative

## Abstract

*Introduction*: With two thirds of adults in paid employment and one third physically inactive, workplaces are an important setting for promoting more physical activity. We explored the attitudes and practices of employees and managers from different industries towards sitting and moving at work, to inform the development of acceptable solutions for encouraging businesses to adopt activity-promoting workplaces. *Method*: We conducted focus groups with employees and structured interviews with upper/middle managers from 12 organisations in a range of industries (e.g., education, healthcare, manufacturing, construction, insurance, mining). Topics focused on past and current workplace health and wellness initiatives, workplace culture and environment related to physical activity, responsibility for employee physical activity patterns at work, and enablers of/barriers to activity promoting workplaces. *Results*: Physical activity was not an explicit priority in existing occupational health and wellness initiatives. Instead, there was a strong focus on education about preventing and managing injuries, such as manual handling among non-office workers and desk-based ergonomics for office workers. Physical activity was viewed as a strategy for maintaining work ability and preventing injury, particularly in blue-collar staff, rather than for chronic disease prevention. Managers noted structural/organisational barriers/enablers to promoting physical activity at work (e.g., regulations, costs, competing concerns), while employees tended to focus on individual constraints such as time and geographic location. The issues of "initiative overload" and making physical activity a part of “business as usual” emerged as strong themes from employees and managers. *Conclusions*: While there is stakeholder enthusiasm for creating activity-promoting workplaces, multi-level support is needed to make physical activity an integral part of day-to-day business. The synergism between occupational health and safety priorities could be leveraged to facilitate the creation of activity-promoting workplaces.

## 1. Introduction

The benefits of physical activity are well-established, yet one third of Australian adults do less than 30 min per week of moderate-to-vigorous physical activity [1]. Both high levels of daily sitting time and prolonged sitting are associated with a greater risk of cardiovascular disease, type 2 diabetes and all-cause mortality, especially among individuals who do little physical activity [2,3].

The workplace is a key setting for delivering health promotion programs to adults [4], with working adults estimated to spend 40% of their waking time at work [5,6]. Workers who are sufficiently active as per guidelines are less likely to take sick leave, have lower presenteeism, and are more productive than those who are physically inactive [7,8]. As the nature of work has become more sedentary and less physically active over the last 50 years across different industries [9], it is important to provide opportunities for workers to move more and be physically active at work.

Activity promoting workplaces are those that facilitate physical activity and/or interrupt prolonged sedentary behaviour. Reviews of workplace interventions to increase physical activity and/or reduce sedentary behaviour have, however, shown inconclusive or equivocal findings, largely due to the diversity of intervention strategies and lack of long-term evidence [10,11]. Facilitators of program success include promoting physical activity through the organisational structure [12,13] and taking a participatory approach with managers and employees to design, plan and implement workplace initiatives [14].

In order to encourage businesses to embrace activity-promoting workplaces more effectively, it is essential to understand the importance placed on physical activity as part of organisational health and safety policies and practices. The aim of this study was to examine Australian manager and employee perspectives on workplace physical activity-related policies and practices.

## 2. Methods

This study was approved by The University of Queensland Human Research Ethics Committee, (No. 2015000186).

### 2.1. Recruitment

Fifty-five workplaces were invited to participate in this study (*n* = 18 in Brisbane, *n* = 37 in Sydney). We made initial contact through ‘cold calls’ to managers based on publicly available information from LinkedIn (*n* = 32) and from researcher team members’ networks and contacts (*n* = 23). Workplaces represented a range of industry sectors (e.g., production, transport, sales, health care) and had a variety of job roles (e.g., shift workers, office workers, drivers, nurses, manufacturing workers, maintenance workers). Managers and employees from participating workplaces were invited to participate in focus groups and/or structured interviews, which were conducted on site at times convenient to the employees. The final study sample involved 12 workplaces (*n* = 4 in Brisbane metropolitan area, *n* = 2 in regional Queensland, *n* = 6 in Sydney metropolitan area) of various sizes and represented ten different industry sectors (see Table 1).

### 2.2. Focus Groups with Employees

Managers promoted the study to their team(s) and 8–10 employees were recruited to participate in each focus group. Participants received written information sheets and provided written consent prior to participation. Each focus group was facilitated by at least one researcher and another researcher took notes.

Focus groups explored the overall workplace culture and practices related to health, safety and wellness; employees’ understanding of prolonged sitting as a potential health issue and whether anything should or could be done about sitting/standing/movement patterns at work; as well as barriers to and facilitators of changing physical activity patterns at work (see Appendix A). Focus groups also explored descriptions of movement patterns during a typical workday and whether these impacted on their recreational activity, but this information is not presented in this paper.

### 2.3. Structured Interviews with Managers

We interviewed at least one upper/middle manager or senior Occupational Health and Safety representative at each participating workplace. These interviews were conducted on the same day as the employee focus groups at a time that suited the manager(s).

Interviews covered similar topics to those in the focus groups and additionally explored managers’ views of activity-promoting workplaces, any activity-promoting policy initiatives that have been introduced, and the barriers to and facilitators of introducing strategies to increase physical activity and reduce sitting in the workplace (see Appendix A).

### 2.4. Analysis

All focus group and interview sessions were audio recorded and transcribed. We analysed focus group and interview transcripts separately using NVivo qualitative data analysis software (Version 11, 2015, QSR International Pty Ltd., Victoria, Australia) and identified the main and subsidiary themes arising from discussions and responses in the focus groups and interviews. Emerging codes were independently identified by one member of the research team (T.K.-A., J.Y.C., L.E., S.Y., H.O.) and verified by a second researcher (T.K.-A., J.Y.C., L.E., S.Y., H.O.). For the purposes of this study, we focused on results about policies and practices related to promoting physical activity and reducing prolonged sitting, and where these initiatives fit within the broader context of workplace health and safety in the organisation.

## 3. Results

The main themes identified were about current workplace health promotion practices in general, and promotion of physical activity specifically; facilitators and barriers to physical activity promotion in the workplace; and the question of individual responsibility versus structural solutions.

### 3.1. Current Workplace Health Promotion Practices (General)

When asked about their current strategies for promoting health in the workplace, physical activity was not an explicit priority at all. Employees and managers mostly referred to initiatives about injury prevention and management, safety, mental health, nutrition, sun safety, smoking cessation, and alcohol consumption.

There was a strong focus on education and provision of information about reducing injury risk and safe manual handling in the trade, mining, construction, manufacturing and healthcare workplaces. For office employees, desk-based ergonomics was the main focus for injury prevention and management.

*From day 1 … [we] talk about the ergonomic principles when people start so that their layout is appropriate and provide workplace and workstation assessments and guidance material, presentations for all the people to run through*.(Employee, Construction)

*We have a healthy work initiative as well which promotes healthy eating, stress management and all those sorts of things. So that’s always up on the noticeboard and different things are there all the time*.(Employee, Education & Training)

### 3.2. Current Workplace Physical Activity Promotion Efforts

When specifically prompted about physical activity initiatives, the majority of responses focused on opportunities and/or facilities for fitness and exercise for employees, such as discounted gym memberships or having external providers deliver on-site classes. There was a tendency to contextualise physical activity as a structured behaviour to be fitted into the workday or before/after work, such as an educational workshop or gym session, rather than talking about incidental activity and interrupting prolonged sitting time during the working day.

Deliberate workplace efforts to promote physical activity were framed as a means to maintain the ability of staff to remain fit for work and avoid injury. These comments related to managing musculoskeletal health and injury risk via stretches and core strength training, and mainly applied to those companies with some ‘blue collar’ staff (e.g., manufacturing, gardening, trade roles). The predominant drivers were proximal and emphasised promoting safe work practices and preventing injuries, rather than preventing chronic disease or promoting wellbeing, which were more distal concerns.

*We sent some off to the gym to improve their core muscles, because what we found out, a lot of the back problems actually came from weakened core muscles, so we paid for gym memberships for those people to do that type of thing*.(Manager, Transport)

*We ran a stretching to action program mainly based for the manufacturing [staff] to warm up prior to their work*.(Employee, Manufacturing)

Community and mass participation events (e.g., the City2Surf fun run, Global Corporate Challenge (GCC)) were mentioned. From this perspective, physical activity was seen as a vehicle for team building, staff engagement, and fundraising for a common cause.

*We’re subsidizing the enrolment for GCC this year, so that means people don’t have to pay full price*.(Manager, Insurance)

*With Australia Day we had a few fun things like thong [flipflop] throwing and egg & spoon races, and things like that*.(Employee, Healthcare)

There was relatively little mention of promoting incidental activity or reducing prolonged sedentary behaviour as part of workplace physical activity initiatives. When probed, initiatives related to “moving more and sitting less” were considered to be novel ideas more relevant to office workers, although there was some awareness among participants in non-office based roles that sitting for prolonged periods was not ideal for other occupations as well. Incidental activity was also framed as part of taking a break, whether from operating machinery or from working at a desk.

*We are starting to see some adopt certain behaviours like a walking meeting or something like that but it’s something that’s still in early stages*.(Employee, Manufacturing)

*They claim that if you stand each day, that it adds another 20 min to your life. So, every day that you are sitting down is shortening your life*.(Employee, Education & training)

### 3.3. Facilitators and Barriers to Promoting Physical Activity at Work: Individual vs. Structural

Employees tended to focus on individual responsibility and personal contexts as influences on their ability to be more physically active during work hours. They referred to the presence/absence or time, personal appeal of initiatives that are implemented, and perceived support from managers, as well as physical location (e.g., being based at different locations to head office), as influences of their physical activity patterns at work.

*The workforce is spread out everywhere. A lot of rural guys out west and coastal guys, so something implemented in [main office] … it doesn’t speak to a large portion of the company*.(Employee, Mining)

*I tried going to the gym while I was doing it. But you only get, really, half an hour, forty-five for lunch*.(Employee, Trade/technical)

*Sometimes there is that attitude that if you’re not seen to be at your desk then you’re not doing anything*.(Employee, Construction)

In contrast, managers tended to focus on structural issues that facilitated or hindered physical activity promotion at work. They frequently cited cost as a barrier to implementing more physical activity-related initiatives. However, managers also talked about being resourceful and taking opportunistic action for developing more activity-friendly facilities.

*It’s not been seen as a priority, because of the cost of it, but I would say that going forward in health you will find that as they do relocation work within facilities and refits within facilities… you may find over the next 10 to 20 years that there won’t be a standard sit-desk anywhere in the facility*.(Manager, Healthcare)

*When we built the office, we put in showers in both wash rooms so that would be available to encourage people to cycle to work or whatever it is that they want to do*.(Manager, Food & beverage)

Overall, however, there was a sense of being inundated with numerous initiatives related to health or other issues in the workplace, which made it difficult to focus on physical activity, even if they wanted to. These managers expressed a sense of being stretched by the different demands and needs of bosses, clients, regulators, and employees. They cited having too many competing priorities, of which physical activity was not the most urgent issue. Meeting government regulations and ensuring safe work conditions took greater precedence, especially among managers responsible for a blue-collar workforce.

*I think our industry is in initiative overload. You’ve got clients trying to push direction on us, you’ve got regulators trying to push direction onto us and I think that’s probably the biggest risk, trying to have too many initiatives*.(Manager, Mining)

*I don’t concentrate on that [evaluation]. I don’t measure their uptake. I don’t say “this many people are on the wellness course or haven’t”. Mainly because I have to do all those risk assessments*.(Manager, Trade/technical)

For companies with a workforce consisting of diverse blue- and white-collar roles and/or spread out in different geographic locations, complying with regulations, as well as delivering different experiences for staff, were difficult to coordinate.

*This industry, in particular, there’s always been a big disconnect between the field and the office. It’s hard because it’s remote locations… they could be in the desert in the middle of nowhere and depending on the leadership on site, it really influences the way they act*.(Manager, Mining)

*It’s really up to us whether we want to do something. Even if we want to do some collective activity to promote ... you know, physical activity at work place, it’s still not that easy considering everyone is playing a different role*.(Manager, Food & Beverage)

According to managers, the challenge was embedding physical activity into usual practices, like those mandated by work health and safety regulations. Nonetheless, managers expressed difficulty in seeing how to incorporate physical activity into everyday business.

*I’m pretty negative on rolling things out as initiatives. I think you’ve got to build it into the business and get to a stage where it’s just what you do as part of the business*.(Manager, Mining)

Overall, employees and managers shared the view that health was the responsibility of both the company and individual workers. There was a sense of joint responsibility and endeavour for health promotion at work, whether it be to increase physical activity or to enhance other aspects of health (e.g., diet, mental wellbeing).

*I think we’re pretty proactive here, like if there’s an issue we’ll raise it up and like, you don’t have to wait until the monthly meeting, we’ll just talk about it and just between us all we tend to come up with some sort of solution about it*.(Manager, Manufacturing)

Managers also noted that, while employee health was in part the company’s responsibility, they respected individual choice; ultimately, employees would make the decision whether or not to engage in health enhancing behaviours. Even if activity-promoting workplace initiatives were in place, employees may not be more active.

*Similarly, I know that there’s a gym here. There’s yoga classes at lunchtime. I think most people are aware of that, but again it’s just personal choice as to whether they choose to access it*.(Manager, Healthcare)

*We can’t force anybody to do anything but it’s up to them*.(Manager, Education & Training)

Finally, having role models for being physically active emerged for both employees and managers as a salient factor. Although many employees believed being physically active was an individual’s responsibility, they acknowledged that it is powerful to have a manager as a role model.

*He (the manager) walks out to see us at our desk, he doesn’t make us go to see him in his room. He’ll walk across to our desk and ask us what we need*.(Employee, Construction)

*If your manager exercises and is active and promotes it in the workplace, you will tend to start doing the same things, because it’ll make it okay, it’ll be all right for you to do those things*.(Employee, Insurance)

Some managers emphasised their own responsibility as role models and champions for change at work and emphasised the importance of top-down leadership. These managers did not see their role as “spoon feeding” health-related information to their employees, but rather leading by example and garnering approval from all levels of the organisation. Their experience showed that programs were best implemented with buy-in and support from senior management and the broader workforce.

*I think it’s got to be led from the top and I think the blokes have got to trust you. The workers have got to trust you and it can’t be, “Ah I’m telling you what to do,” sort of thing, it’s just got to be led by example without too much telling them and then they just pick up and start doing*.(Manager, Manufacturing)

*We did actually have an incentive and we got buy in from our manager, the director of workforce and culture, to provide membership for one year* (for the fitness passport) … *we had to actually get 12 hundred people joining up within a certain period of time.*
(Manager, Healthcare)

*I think leadership’s a big part of it, your leaders got to be seen to be out doing this stuff. Our CEO, she goes out on every Saturday and has a trainer. Our CFO runs to work every day*.(Manager, Transport)

## 4. Discussion

This study provides insights into Australian manager and employee perspectives about workplace physical activity-related policies and practices across a range of industry sectors. Our results show that these Australian workplaces did not prioritise physical activity promotion explicitly, although in some cases, it was embedded as part of other occupational health and safety practices, such as injury prevention. Where there were examples of physical activity promotion, they mostly focused on the provision of structured exercise opportunities during and outside worktime rather than on incidental movement and breaking up prolonged periods of sitting. Overall, these findings highlight a need to reframe the important role of workplace physical activity promotion in overall health promotion and injury prevention strategies.

Our participants suggested that possible activity-promoting solutions at work would involve participatory, multicomponent, and holistic approaches. This is consistent with the literature on physical activity promotion and on workplace health promotion more generally [14,15,16]. Participants also noted a tension between individual choice and company responsibility; employers might provide physical activity opportunities, but participation is ultimately up to the employees. In the UK, employers have also voiced concerns about balancing the desire to implement compulsory physical activity or sitting breaks with employees’ personal choice [17]. Nonetheless, the view that employers should not interfere with employees’ health behaviours and private lives [18] appears to be changing, with a move to employers facilitating healthy behaviours rather than enforcing certain practices, and for employees to be more than passive program recipients. We found an overarching view from participants that promoting physical activity (and any health-related initiatives) through the workplace is a joint endeavour, with responsibility at all levels of management and with employees themselves. Similar views have been expressed before by office workers [19,20], and this study demonstrates the salience of this view across industry sectors, many of which do not involve offices or desk-based roles (e.g., nursing, cleaning, retail, manufacturing, construction).

The results of this study highlight the challenge of embedding physical activity promotion in regular work practice. How can it become a part of business as usual? This problem is echoed by other researchers [17,21] and potential solutions are multifaceted. In their systematic review of Total Worker Health Interventions, Feltner et al. [14] identified the participatory approach (e.g., creation of a joint management-employee advisory group) to developing, designing, planning, and implementing programs, as a key element of success. The merits of participatory approaches to co-development of workplace interventions have been demonstrated in formative ‘move more, sit less’ studies [22,23,24]. Additionally, multicomponent initiatives are more likely to be efficacious than single-focus ones [14,25], with greater potential for sustainability [12]. Karanika-Murray and Weyman [15] advocate for a more holistic approach and suggest moving away from the current hierarchical prioritisation of risk reduction and management in workplace health. They talk of changing to focus on the workplace as a resource for promoting better health and wellbeing rather than as a source of health risks and poor health, reiterating the position of the World Health Organization and World Economic Forum [4] in the workplace health management sector. Physical activity fits well within this conceptualisation, given its well-known associations with physical and mental health [26,27]. Yet, reviews about workplace health promotion and physical activity interventions suggest that integrated and holistic approaches are relatively rare [10,11,14,16], and indicate future directions for this field.

Furthermore, diversity in workforce characteristics (e.g., physical requirements and/or location) poses a challenge to effective physical activity promotion and uptake of initiatives. Analysis by industry sector has highlighted the challenge of implementing workplace-based health promotion programs, due to the mix of positions and job roles (e.g., blue and white collar; managerial, clerical, and technical staff) within each workplace and within each industry or sector [28]. Focusing health promotion efforts on one group could inadvertently exclude or discourage efforts to target other groups, which is neither appropriate nor equitable. This issue was also noted by participants in our study. Targeting large organisations based in metropolitan regions would achieve the greatest reach, but the highest proportion of workers with multiple health risks are located in non-metropolitan areas and in industries with a high proportion of small businesses [28].

### 4.1. Implications of These Findings

Altogether, our results suggest that ways to embed physical activity into business as usual are needed. Since physical activity has established benefits for a range of health issues (e.g., stress management, overweight/obesity, musculoskeletal pain, and injury prevention), encouraging activity-promoting workplaces could simultaneously address multiple occupational health-related concerns. As mentioned above, workplaces are already overloaded with initiatives, and any activity-promoting initiatives will require engaged participation from employees and managers.

In the past, smokers were permitted to take regular breaks during their work day or shift. Vis-à-vis, perhaps it is time for workplaces to permit regular activity breaks? One or two pre-scheduled 15-min walks or light strength training during work time, plus more opportunities for incidental activity, could help employees to meet the current physical activity guidelines on their work days.

Existing knowledge about what works in promoting physical activity at work could be sourced from the evidence base and curated based on the needs of employees’ job roles and organisational context. This knowledge could then be used to form the basis of the participatory process through which workplaces can consult internally and design their own tailored initiatives [14,16]. Solutions may take the form of designated activity breaks during the workday (e.g., ‘Booster Breaks’ [29]) or the incorporation of more incidental activity throughout the workday, alongside existing occupational health and safety practices [22,24]. The feasibility of developing tailored strategies for activity promoting workplaces has, however, rarely been tested in diverse industry sectors. As such, the development and evaluation of activity promoting strategies in a range of workforces, and testing their generalisability across different industries, could be a priority for future research.

### 4.2. Strengths and Limitations

The main strength of this study is the diversity of industry sectors, and management and employee roles among the participants, increasing its generalisability across different workforces and countries. Other studies of this kind have involved mostly white-collar organisations or workers. For example, Knox et al. [21] surveyed 3360 adults working in 308 English workplaces; of the respondents, 97% had managerial, professional, or clerical/administrative positions. A limitation of our study is that there may have been selection bias towards more activity-friendly companies, as the stated purpose of the research at recruitment was to explore factors that would contribute to creating activity-promoting workplaces. However, participants were encouraged to express positive and negative sentiments about all issues discussed. Employees were also not aware that physical activity was the main outcome of interest as the questions were framed to focus on work and health, and not exclusively about physical activity and sedentary behaviour at work.

## 5. Conclusions

Different and innovative ways to promote and embed physical activity at work are needed to address this public health issue. Current practices, such as core strength training and stretch breaks, that reflect occupational health and safety regulations, provide some opportunities for physical activity during the workday, but other initiatives generally rely on the buy-in and dedication of managers, many of whom are “in initiative overload”. Given the beneficial links of physical activity with other issues addressed by workplace health and wellness initiatives, there is an opportunity to take synergistic action that can provide benefits for individuals and their workplaces.

## Figures and Tables

**Table 1 ijerph-16-00516-t001:** Industry sector and occupations of study participants.

Workplace ID	Sector	*N*	Employee Job Roles (*n*)	Manager Roles (*n*)
1	Education and training	9	Trainers (6); Industry mentor (1); Project manager (1)	Managing Director (1)
2	Healthcare	10	Speech therapists (5); Corporate administrative officers (4)	Human Resources (1)
3	Trade/technical	4	Gardeners (2); Project manager (1)	Occupational Health & Safety (1)
4	Manufacturing	6	Steel fabricators / Welders (4); Project manager (1)	Co-owner and Office Manager (1)
5	Transport	13	Runway safety officers (12)	Human Resources (1)
6	Mining	12	Rig crew (3); Camp chef (1); Cleaner (1); Warehouse and workshop personnel (3); Field safety officer (1); Pilot (1); Aircraft engineer (1)	Occupational Health & Safety (1)
7	Construction	8	Corporate staff /Project managers (7)	Occupational Health & Safety (1)
8	Insurance	8	Claims staff (4)	Human resources (4)
9	Food and Beverage	13	Sales staff (3); Administrative officers (7)	General Manager (1); Human Resources (1); Marketing (1)
10	Healthcare	12	Nurses (3); Administrative assistants (3); Cleaners (2); Patient transport officers (2); Food service assistants (1)	Human Resources (1)
11	Manufacturing	11	Office workers (5); Production workers (2); Other site managers (3)	Occupational Health & Safety (1)
12	Healthcare	8	Merit clinicians (2); Nurse (1); Personal assistant (1); Research assistant (1); Counselling clinician (1)	Department Manager (2)

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
