# Peer review of "“In Initiative Overload”: Australian Perspectives on Promoting Physical Activity in the Workplace from Diverse Industries"

_ijerph, 2019, doi:10.3390/ijerph16030516_

Reviewer 1 Report

In the Introduction:

The authors wrote “Healthy workers are less likely to take sick leave, have lower presenteeism, and are more productive than those who are physically inactive”. This part of the text presents some problems: 1) it seems to consider "physical inactivity" and "physical inactivity in leisure" as synonyms; 2) considers that "physically inactive people" are not healthy; 3) establish a causal relationship between physical inactivity (in leisure? At work? Or both?) and take sick leave, presenteeism and productivity, when perhaps healthier people should be the same ones who do more physical activity, are more productive and, of course, less likely to take sick leave.

The authors wrote “As the nature of work has become more sedentary and less physically active over the last 50 years across different industries, it is important to provide opportunities for workers to move more and be physically active at work”. I believe that people should be more active at leisure and not, exactly, at work.

The authors wrote “These employee behaviours are associated with benefits to the organisation in terms of absenteeism, staff turnover, morale and wellbeing, engagement, productivity, and positive attitudes towards the workplace”. This part of the text presents "value judgments".

In the Conclusions:

The authors wrote “Significant proportions of working adults are not sufficiently active for good health”. This phrase is dispensable and could be removed.

Author Response

In the Introduction:

R1.1.      The authors wrote “Healthy workers are less likely to take sick leave,

have lower presenteeism, and are more productive than those who are

physically inactive”. This part of the text presents some problems: 1) it

seems to consider "physical inactivity" and "physical inactivity in leisure" as

synonyms; 2) considers that "physically inactive people" are not healthy; 3)

establish a causal relationship between physical inactivity (in leisure? At

work? Or both?) and take sick leave, presenteeism and productivity, when

perhaps healthier people should be the same ones who do more physical

activity, are more productive and, of course, less likely to take sick leave.

A1.1.      We have clarified this sentence as follows, “Workers who are sufficiently active as per guidelines are less likely to take sick leave, have lower presenteeism, and are more productive than those who are physically inactive [7,8].”

R1.2.      The authors wrote “As the nature of work has become more

sedentary and less physically active over the last 50 years across different

industries, it is important to provide opportunities for workers to move more

and be physically active at work”. I believe that people should be more active

at leisure and not, exactly, at work.

A1.2.     We agree with this Reviewer that the benefits of leisure time physical activity are well established. However, the benefits of other domains of physical activity (e.g., occupational, transport, household) for reducing risk of chronic ill health and mortality has also been demonstrated. For example, see Samitz et al (2011)’s systematic review and dose-response meta-analysis that showed associations of higher levels of total and domain-specific physical activity with lower risk of all-cause mortality.

Reference:  Samitz G, Egger M, Zwahlen M. Domains of physical activity and all-cause mortality: systematic review and dose–response meta-analysis of cohort studies. International journal of epidemiology. 2011;40(5):1382-400. https://academic.oup.com/ije/article/40/5/1382/658632

R1.3.      The authors wrote “These employee behaviours are associated with

benefits to the organisation in terms of absenteeism, staff turnover, morale

and wellbeing, engagement, productivity, and positive attitudes towards the

workplace”. This part of the text presents "value judgments".

A1.3.      We have removed this statement as it repeats information statement made in paragraph 1, “Healthy workers are less likely to take sick leave, have lower presenteeism, and are more productive than those who are physically inactive [7,8]”

In the Conclusions:

R1.4.      The authors wrote “Significant proportions of working adults are not

sufficiently active for good health”. This phrase is dispensable and could be

removed.

A1.4       This phrase has been deleted to reduce repetition as suggested.

Reviewer 2 Report

This study is dealing with the attitudes and practices of employees and managers from different industries towards sitting and moving at work, to inform the development of acceptable solutions for encouraging businesses to adopt activity-promoting workplaces. For this purpose, the authors conducted focus groups and interviews with employees and managers of 12 organisations in a range of industries.

This study provides very interesting results for physical activity promotion in workplaces and adds up to an important conclusion.

I only want to address some minor comments for revision:

-          L.54: can you please include a reference here

-          Table 1: maybe you can differentiate participants of focus groups into two columns (employees/managers) and then describe them

-          Could you please give more information on the structured interview? What questions were asked?

-          Comprehension questions: were the managers “just” managers or were they responsible for the occupational health and safety in their company? (l. 89)

-          What were your reasons to analyse and present the results of the focus groups and structured interviews together? Would it not also make sense to analyse and present the results for each group separately?

-          There are some space characters to many in the manuscript

Author Response

This study is dealing with the attitudes and practices of employees and

managers from different industries towards sitting and moving at work, to

inform the development of acceptable solutions for encouraging businesses

to adopt activity-promoting workplaces. For this purpose, the authors

conducted focus groups and interviews with employees and managers of 12

organisations in a range of industries.

This study provides very interesting results for physical activity promotion in

workplaces and adds up to an important conclusion.

I only want to address some minor comments for revision:

R2.1.       - L.54: can you please include a reference here

A2.1.      Please refer to response A1.3 above. We removed this sentence to reduce repetition.

R2.2.       - Table 1: maybe you can differentiate participants of focus

groups into two columns (employees/managers) and then describe

them

A2.2.       Table 1 has been revised as suggested to show employees and managers separately.

R2.3.       - Could you please give more information on the structured

interview? What questions were asked?

A2.3.      The interview guide is now included as supplementary materials. We have also revised the manuscript to say that some topics are not presented in this paper as follows, “Focus groups also explored descriptions of movement patterns during a typical workday and whether these impacted on their recreational activity, but this information is not presented in this paper.” [Focus groups with employees, last sentence]

R2.4.      - Comprehension questions: were the managers “just”

managers or were they responsible for the occupational health and

safety in their company? (l. 89)

A2.4.      We have provided information about the type of managers interviewed as follows in the manuscript, “We interviewed at least one upper/middle manager or senior Occupational Health and Safety representative at each participating workplace” [Structured interviews with managers, line 1]

R2.5 - What were your reasons to analyse and present the results of

the focus groups and structured interviews together? Would it not

also make sense to analyse and present the results for each group

separately?

A2.5.     The data for focus groups and interviews were analysed separately. We have presented the findings together to provide a more inclusive picture of the results and to show similarities and/or differences between employees and managers side-by-side. To make this clearer the manuscript has been revised as follows: “We analysed focus group and interview transcripts separately using QSR International NVivo (Version 11) and identified the main and subsidiary themes arising from discussions and responses in the focus groups and interviews.” [Analysis, line 1]

R2.6. - There are some space characters to many in the manuscript

A2.6.      Space characters have been removed

Reviewer 3 Report

Itwould be interesting for the authors to develop the issue of sustainable management of proactive behavior in the workplace. I mean, maybe people who are saturated with physical activity even at a low level in the workplace may not be able to miss sport in their free time, which may have negative longterm effects. For example, in Central and Eastern Europe where the promotion of physical activity in the workplace is not so popular, there is a dynamic turn towards physical activity in free time.

For example, Physical activity of Poles has increased dynamically in the last two decades.

Positive changes began to be noticed after the political changes in Poland in 1989.

Earlier, Poles were a community that, contrary to Western societies, showed much lower physical activity. After Poland’s accession to the European Union, Poland was at the end of the list of European countries in terms of physical activity. It is currently in the middle of the list.

It is a rapid growth, and now the media and politicians are trying to consolidate this trend. Social, cultural and economic factors influenced the increase of physical activity of Poles. Currently, Poles are better educated, wealthier, have more free time. Poles have moved to cities in which sports infrastructure has developed dynamically-swimming pools, fitness clubs and bicycle paths. THE POLES ALSO MOVED TO THE OFFICES - Earlier, they did a lot of physical or agricultural work. That is why they no longer thought about running or swimming...

Author Response

It would be interesting for the authors to develop the issue of sustainable man

agement of proactive behavior in the workplace. I mean, maybe people who

are saturated with physical activity even at a low level in the workplace may

not be able to miss sport in their free time, which may have negative long term

effects. For example, in Central and Eastern Europe where the promotion

of physical activity in the workplace is not so popular, there is a dynamic turn

towards physical activity in free time.

For example, Physical activity of Poles has increased dynamically in the last

two decades.

Positive changes began to be noticed after the political changes in Poland in 1989.

Earlier, Poles were a community that, contrary to Western societies, showed

much lower physical activity. After Poland’s accession to the European

Union, Poland was at the end of the list of European countries in terms of

physical activity. It is currently in the middle of the list.

It is a rapid growth, and now the media and politicians are trying to

consolidate this trend. Social, cultural and economic factors influenced the

increase of physical activity of Poles. Currently, Poles are better educated,

wealthier, have more free time. Poles have moved to cities in which sports

infrastructure has developed dynamically-swimming pools, fitness clubs and

bicycle paths. THE POLES ALSO MOVED TO THE OFFICES - Earlier, they

did a lot of physical or agricultural work. That is why they no longer thought

about running or swimming...

RESPONSE TO REVIEWER 3: Thank you to Reviewer 3 for sharing their insights about the changes in physical activity in Poland in the last few decades. Our study describes perspectives from a range of industries and workplaces in Australia, and findings could be generalisable to workplaces in other countries. We revised the manuscript to reflect this as follows, “The main strength of this study is the diversity of industry sectors, management and employee roles among the participants, increasing its generalisability across different workforces and countries.”[Page 15, paragraph 1, line 1]

Reviewer 4 Report

The results are not quantified and the evaluations are approximate and subjective.  For example:  - the themes identified were ...  - Employees and managers  referred ....

- There was a strong focus on......

- the main focus for ...

- the majority of responses

- etc.

Author Response

The results are not quantified and the evaluations are approximate and subjective.

For example: - the themes identified were ...

- Employees and managers referred ....

- There was a strong focus on......

- the main focus for ...

- the majority of responses

- etc.

RESPONSE TO REVIEWER: Thank you to Reviewer 4 for their suggestions regarding presentation of the results. We acknowledge that there are various ways to present these qualitative results. As this manuscript has been reviewed and revised iteratively by myself and my co-authors, we feel it is acceptable and clear in its current format. The results are discussed by themes and we have used expressions similar to that suggested by Reviewer 4 above.